# Enhancing Corporate Sustainability through Just-In-Time (JIT) Practices: A Meta-Analytic Examination of Financial Performance Outcomes

Javier García-Cutrín [1,*] and Carlos Rodríguez-García [2]

1 Departament of Mathematics, ECOBAS, University of Vigo, 36310 Vigo, Spain
2 Departament of Business Organization, ECOBAS, University of Vigo, 36310 Vigo, Spain; c.rodriguez@uvigo.gal
* Correspondence: fjgarcia@uvigo.gal

**Abstract:** This paper utilizes a meta-analytic approach to examine the correlation between Just-In-Time (JIT) practices and financial performance. The investigation assesses JIT's influence on key financial metrics, including Return on Investment (ROI), Return on Sales (ROS), Asset Turnover, and Profit Margins. Results indicate a robust positive correlation between JIT implementation and enhanced financial outcomes, demonstrating that JIT significantly contributes to both operational efficiency and financial health. The findings illustrate that JIT's effectiveness varies by organizational scale and economic context, with larger firms and stable economic conditions seeing the most pronounced benefits. Conversely, smaller firms might face challenges in harnessing JIT's full financial potential, underscoring the importance of tailored implementation strategies. This study confirms the strategic value of JIT for boosting profitability and efficiency, providing managers with actionable insights to optimize JIT deployment. It also suggests areas for future research to refine understanding of JIT's variable effects across different operational contexts and explore potential contributions to sustainability initiatives. This contribution enriches the discourse on JIT's role in enhancing corporate financial performance and opens the door to examining its broader impacts on sustainable business practices.

**Keywords:** Just-In-Time (JIT); financial performance; operational excellence; lean manufacturing; Toyota Production System (TPS); meta-analysis

## 1. Introduction

The adoption of Just-In-Time (JIT) systems has revolutionized manufacturing processes over recent decades, enhancing operational efficiency and providing a basis for corporate sustainability in the context of long-term financial growth [1–3]. Originating from the Toyota Production System, the essence of JIT—reducing waste and enhancing production flow—has been universally celebrated for its operational merits [4–6]. Yet, the challenge of translating these operational advantages into measurable financial outcomes and assessing their contribution to sustainable economic growth remains largely unexplored [7–9].

While the operational advantages of JIT, such as inventory reduction and improved throughput, are well-documented [10], the exploration of JIT's broader financial implications, including its impact on profitability, return on investment, and the overall financial well-being of organizations, requires further attention [8]. This gap is particularly pressing considering the growing expectations for manufacturing firms to not only achieve operational excellence but also to ensure financial viability over the long term [11,12].

This study seeks to bridge this critical gap by systematically examining the financial outcomes of JIT implementation across a spectrum of manufacturing firms. Employing a comprehensive analytical framework, this research aims to elucidate the relationship between JIT operational practices and financial performance indicators, with a focus on long-

term corporate sustainability. The investigation, guided by the Problem, Intervention, Outcome (PIO) framework, hypothesizes that "Manufacturing firms employing JIT in their operations exhibit significant improvements in financial performance post-implementation", a proposition explored through both quantitative and qualitative analyses.

Contributing to the literature in two significant ways, this study theoretically extends the understanding of JIT by connecting operational efficiencies to financial sustainability. Practically, it provides managers and decision-makers with evidence-based insights into the long-term financial benefits of JIT adoption, supporting strategic planning and implementation efforts aimed at achieving enduring financial stability. By highlighting the conditions under which JIT implementation is most financially and environmentally beneficial, this research contributes to a nuanced understanding of JIT as a strategic tool for enhancing both operational efficiency and sustainability in financial performance.

The paper is structured as follows. Following this introduction, Section 2 reviews the relevant literature on JIT and its operational and financial implications. Section 3 outlines the research methodology, including the analytical framework and data collection process. Section 4 presents the findings, and Section 5 discusses the theoretical and practical implications of the study and concludes with a summary of the key contributions and suggestions for future research.

## 2. Literature Review

This literature review critically evaluates the extensive research on Just-In-Time (JIT) manufacturing, its impact on operational efficiency, and the subsequent financial performance outcomes, underscoring the necessity for a meta-analysis to synthesize fragmented insights.

JIT, originating from the Toyota Production System [4], revolutionized manufacturing by emphasizing waste reduction and the seamless flow of goods. While some define JIT as a managerial philosophy, others see it as a collection of specific practices, indicating a broad spectrum of implementations and conceptual understandings within the field [13,14]. Seminal works in the late 20th century, including those by Monden [5] and Shingo [15], have significantly shaped JIT's research directions, transitioning from practical applications to broader considerations including purchasing and logistics. Numerous studies have confirmed JIT's efficacy in enhancing operational metrics, such as inventory levels, production lead times, and quality control, establishing a solid foundation for its operational advantages [1,16].

While the operational benefits of JIT are well-documented, the translation of these efficiencies into tangible financial performance has presented a more complex scenario [10]. Initial theories proposed that operational improvements should naturally culminate in financial betterment through cost reduction and heightened productivity [1,17]. Nonetheless, empirical evidence has been mixed, with some studies reporting significant financial gains post-JIT implementation (e.g., Cua et al. [18]; Mackelprang and Nair [10]; Fullerton et al. [19]), while others highlight contingencies based on firm size, industry sector, and implementation depth [20,21].

Emerging research has attempted to bridge these gaps, investigating how JIT's operational benefits convert into financial success. Factors such as supply chain integration, customer satisfaction, and market responsiveness have been identified as crucial for leveraging JIT's efficiencies for financial improvement (e.g., Swink et al. [22]; Valente et al. [20]). Furthermore, the alignment of JIT practices with new technologies, broader organizational strategies, and market conditions is shown to significantly influence financial outcomes [23–25].

Recent literature also suggests a potential linkage between JIT practices and environmental sustainability. The inherent focus of JIT on reducing waste and improving resource efficiency aligns with broader sustainable business objectives, as demonstrated by some studies [26], which explore how operational efficiencies under JIT can support environmental and social goals. This insight invites consideration of JIT not only as a financial tool but

also as a component of sustainable operational strategies, although this paper primarily focuses on the financial implications.

The existing body of JIT literature, predominantly characterized by case studies and cross-sectional surveys, underscores the methodological diversity within this research area. However, this diversity also signals a critical gap—the need for a comprehensive meta-analysis that synthesizes these disparate findings into a cohesive understanding. The call for longitudinal studies and sector-specific investigations further emphasizes the importance of a meta-analytical approach to discern the long-term financial impacts of JIT and its applicability across various manufacturing settings [27]. This literature review, therefore, not only highlights the multifaceted impact of JIT on manufacturing operations and financial performance but also firmly establishes the need for a meta-analysis to consolidate existing research, clarify mixed findings, and guide future JIT implementations towards sustainable financial success.

## 3. Methods

This study employs a comprehensive meta-analysis to evaluate the effect of Just-In-Time (JIT) implementation on the financial performance of manufacturing firms. This methodological approach was chosen to synthesize empirical findings across a broad spectrum of studies, addressing the variability in results and interpretations that have characterized previous research in this area [28,29].

Meta-analysis is a type of quantitative literature review that essentially seeks two goals [30,31]: (a) the integration of primary research results by contrasting hypotheses and (b) the presentation of new hypotheses not included in primary research. Thus, in addition to the integration of results or the refutation of established hypotheses, heterogeneity analysis of primary studies opens up new avenues of inquiry [29,32].

Our investigation was motivated by the need to clarify the financial repercussions of adopting JIT practices—a cornerstone of the Toyota Production System. While JIT's operational benefits are well-documented, its direct impact on financial outcomes remains ambiguous [20,21]. We aimed to resolve this by examining whether JIT implementation leads to improved financial performance. The hypothesis posits that manufacturing firms employing JIT practices experience significant financial performance improvements post-implementation.

We included empirical studies that directly examine the relationship between JIT practices and financial performance. Our search was intentionally broad to encompass a diverse array of JIT practices and financial performance metrics (see Table 1), thereby mitigating selection bias. The criteria for inclusion were empirically based quantitative studies published between 1995 and 2024, in peer-reviewed journals covering operations management, logistics, marketing, and related fields. Conceptual, analytical, and case study research were excluded to maintain a focus on empirical evidence [33].

It is also worth noting that the selection of empirical studies spanning from 1995 to 2024 was intentional, reflecting the evolution of Just-In-Time (JIT) practices across various technological and economic landscapes. This period encapsulates significant advancements in manufacturing technologies, supply chain management systems, and global economic fluctuations, providing a comprehensive view of JIT's adaptability and relevance. By examining studies across these years, our analysis aims to capture the sustained impact and transformation of JIT practices in response to these shifts. Furthermore, this broad temporal range allows us to explore the consistency of JIT benefits under varying conditions and to identify trends that may inform future JIT implementations in evolving technological and economic contexts.

**Table 1.** Main terms used in the search process.

| Just-In-Time | Financial Performance |
|---|---|
| Just In Time (JIT) | Financial performance |
| Kanban | Business performance |
| Pull system | Firm performance |
| Takt time | Competitive Advantage |
| Setup time reduction | Earnings/Sales growth |
| Small lot sizes | Return On Equity (ROE) |
| JIT delivery (costumer and client) | Return on Investment (ROI) |
| Daily schedule adherence | Sales growth |
| Unexpected stoppages in production | Revenue growth |
| Preventive maintenance | Profit margin |
| Use of cellular manufacturing design | Productivity |
| Continuous flow (one-piece-flow) | Market Performance |
| Repetitive nature of master schedule | Manufacturing Cost |
|  | Labor productivity |
|  | Manufacturing performance |

Source: Author's own.

A Boolean search was conducted using combinations of dependent and independent variables (e.g., "Just In Time" + "Financial Performance"; see Table 1). The search spanned several databases with terms appearing in the title, keywords, and abstract. Additionally, we extended our search to specific journals renowned for their contributions to operations management and related disciplines (e.g., Academy of Management Journal, Journal of Operations Management, Strategic Management Journal) to ensure comprehensive coverage of relevant literature.

The first selection was refined by eliminating duplicate articles and eliminating those not written in English. To further ensure objectivity, only papers published in indexed WoS and SJR journals were included. Thus, conference reviews, book reviews, book chapters, and undefined research papers were excluded to maintain the quality of the study (see Table 2 for the final list of journals). Finally, we used an inter-rate reliability test to eliminate any articles that did not fit our search objectives. The key criteria for inclusion were that JIT or Lean had to appear as the centerpiece of the article and that it had to be related to financial performance. To maximize the robustness of the study, the authors independently analyzed the selected articles, and the findings were subsequently compared to assess possible differences.

From an initial pool, twelve studies met our criteria for inclusion (Table 3). This selection process involved rigorous screening for empirical evidence linking JIT practices to financial outcomes, with a preference for studies offering clear, quantifiable measures of financial performance [34]. Although there is no minimum number of articles that must be included in a meta-analysis, methodological guidelines and reference experts suggest considering at least 10 studies to be able to perform adequate statistical analyses and obtain meaningful conclusions [31,32].

Data were extracted from the selected studies, therefore focusing on reported correlations between JIT practices and financial performance indicators [35]. In cases where direct correlations were not reported, regression coefficients were used to estimate the effect sizes, adhering to established statistical methods for converting $\beta$ coefficients to correlation coefficients ($r$).

**Table 2.** Journals in which studies were searched.

| Name of the Journal |
|---|
| Academy of Management Journal |
| Academy of Management Review |
| Administrative Science Quarterly |
| Decision Sciences |
| IEEE Transactions on Engineering Management |
| International Journal of Logistics Management |
| International Journal of Operations & Production Management |
| International Journal of Physical Distribution and Logistics Management |
| International Journal of Production Economics |
| International Journal of Production Research |
| International Journal of Purchasing and Materials Management |
| International Journal of Quality and Reliability Management |
| Journal of Business Logistics |
| Journal of Marketing |
| Journal of Marketing Research |
| Journal of Operations Management |
| Journal of Supply Chain Management |
| Management Science |
| Manufacturing and Service Operations Management |
| Operations Management Research |
| Production and Operations Management |
| Strategic Management Journal |
| Total Quality Management |

Source: Author's own.

**Table 3.** Studies selected for the meta-analysis.

| Title | Authors | Year |
|---|---|---|
| The relationship between JIT manufacturing and performance in Mexican plants affiliated with U.S. companies [36]. | Lawrence and Hottenstein. | 1995 |
| Supply-Based Strategies, Human Resource Initiatives, Procurement Leadtime, and Firm Performance [37]. | Jayaram and Vickery. | 1998 |
| The effect of Just-In-Time with Customers on Organizational Design and Performance [38]. | Claycomb et al. | 1999 |
| Just-in-time: A cross-sectional plant analysis [39]. | Callen et al. | 2000 |
| The production performance benefits from JIT implementation [40]. | Fullerton and McWatters. | 2001 |
| Lean manufacturing: context, practice bundles, and performance [14]. | Shah and Ward. | 2003 |
| The impact of Organizational Culture on Time-Based Manufacturing and Performance [41]. | Nahm et al. | 2004 |
| Impact of technological, organizational, and human resource investments on employee and manufacturing performance: Australian and New Zealand evidence [42]. | Challis et al. | 2005 |
| Matching plant flexibility and supplier flexibility: Lessons from small suppliers of U.S. manufacturing plants in India [43]. | Avittathur and Swamidass. | 2007 |
| Lean manufacturing, non-financial performance measures, and financial performance [44]. | Fullerton and Wempe. | 2009 |
| Agile manufacturing: Relation to JIT, operational performance, and firm performance [45]. | Inman et al. | 2011 |
| The relationship of operational innovation and financial performance—A critical perspective [46]. | Klingenberg et al. | 2013 |

Source: Author's own.

From these twelve publications, data were directly extracted from nine publications. In the remaining three, correlations are not directly specified, but two of them include regression models incorporating the study variables, from which $r$ can be estimated. For this reason, the publication by Callen et al. [39] cannot be used in the MA because it does not have standardized $\beta$ coefficients.

On the other hand, correlations can be extracted from the publication by Fullerton and McWatters [40], and the study by Shah and Ward [14] using the Peterson and Brown approximation, which allows for the estimation of $r$ from standardized $\beta$ coefficients. The formula for calculation is as follows:

$$r = 0.98\beta + 0.05\lambda$$

The variable $\lambda$ takes the value of 1 when $\beta$ is positive and the value of 0 when $\beta$ is negative. This approximation is valid when $\beta$ is within the $\pm0.5$ interval, a criterion met in both studies. Despite being able to obtain $r$ in both studies, they cannot both be used because the study by Shah and Ward [14] comes from a multivariate regression. The correlation from the multivariate study is affected by the rest of the variables in the model, making it not comparable with the rest of the obtained correlations. If correlations between predictor variables were available, the tracing rule method could be used. With the tracing rule, the effect of the predictors can be isolated, and any coefficient extracted from a multivariate model, whether regression or structural equations, etc., could be included [47].

Table 4 shows the correlations extracted from the studies individually (correlation column), the study's sample size ($n$), and the code with which we name each study individually (ID). Notice that some of the papers offer correlations of JIT practices with several performance measures, which we indicate in the same table.

This preliminary analysis not only reinforces the recognized benefits of JIT but also highlights nuanced variations in how these benefits manifest under different conditions. To begin with, it is worth noting that the studies employ diverse financial metrics such as Return on Investment (ROI), Return on Sales (ROS), Asset Turnover, and Profit Margins. Each metric provides insights into different facets of financial performance influenced by JIT. For instance, ROI and ROS often measure the profitability impacts, reflecting JIT's effectiveness in reducing costs and enhancing operational efficiency. Asset Turnover indicates how well JIT practices optimize the use of assets, crucial for organizations aiming to maximize their resource utilization. Profit Margins reveal the ability of JIT to improve cost structures through waste reduction and process improvements.

Notably, studies like Shah and Ward [14], which reported a high correlation between JIT practices and labor productivity ($r = 0.472$), suggest that JIT's impact is more pronounced in settings with extensive operational scales. Similarly, Klingenberg et al. [46] showed a significant positive correlation ($r = 0.66$) between inventory turnover and asset turnover, underlining JIT's capacity to enhance asset efficiency. In contrast, studies like Avittathur and Swamidass [43] observed a negative correlation ($r = -0.271$) between JIT practices for small suppliers and profitability, pointing to challenges that smaller firms may face in leveraging JIT effectively.

The variability in the strength and direction of these correlations can often be attributed to external factors such as economic conditions, industry characteristics, and market dynamics. For instance, economic downturns or booms can significantly influence how effectively JIT practices translate into financial gains, as seen in sectors like automotive manufacturing, where supply chain disruptions can amplify or mute JIT's benefits. Also, changes in trade policies or manufacturing regulations can also impact JIT effectiveness, as compliance costs and operational adjustments may temporarily disrupt the established JIT benefits. Finally, environmental factors such as shifts in consumer demand, technological advancements, or competitive pressures often necessitate adjustments in JIT implementations, which may explain the fluctuations in financial performance outcomes across different studies.

**Table 4.** Raw correlations between JIT practices and performance measures.

| Publication | ID | $n$ | Correlation | $r_c$ |
|---|---|---|---|---|
| **Lawrence and Hottenstein [36]** | | | | |
| JIT construct with productivity | 1 | 116 | 0.42 | 0.4215 |
| **Jayaram and Vickery [37]** | | | | |
| JIT Manufacturing with ROI | 2 | 49 | 0.118 | 0.1193 |
| JIT Purchasing with ROI | 2 | 49 | 0.199 | 0.2011 |
| **Claycomb et al. [38]** | | | | |
| Clients-JIT construct with financial performance | 3 | 200 | 0.16 | 0.1604 |
| **Fullerton and McWatters [40]** | | | | |
| Set up time reduction with return on sales | 4 | 91 | 0.262 | 0.2634 |
| Cellular manufacturing with return on sales | 4 | 91 | 0.318 | 0.3196 |
| **Shah and Ward [14]** | | | | |
| JIT construct with labor productivity | 5 | 1508 | 0.472 | 0.4721 |
| **Nahm et al. [41]** | | | | |
| Time-based manufacturing construct with a composite performance indicator based on self-reported sales growth, return on investment, market share gain, and overall competitive position | 6 | 224 | 0.435 | 0.4358 |
| **Challis et al. [42]** | | | | |
| JIT construct with manufacturing performance (Likert type scales) | 7 | 1024 | 0.135 | 0.1351 |
| JIT construct with employee performance (Likert type scales) | 7 | 1024 | 0.226 | 0.2261 |
| **Avittathur and Swamidass [43]** | | | | |
| JIT construct for small suppliers with profitability (2 Liker-type scales) | 8 | 26 | −0.271 | −0.2765 |
| **Fullerton and Wempe [44]** | | | | |
| Set up time reduction with ROS | 9 | 121 | 0.22 | 0.2209 |
| Cellular manufacturing with ROS | 9 | 121 | 0.27 | 0.2711 |
| **Inman et al. [45]** | | | | |
| JIT-Production construct with financial performance construct | 10 | 96 | 0.206 | 0.2071 |
| JIT-Purchasing construct with financial performance construct | 10 | 96 | 0.134 | 0.1347 |
| **Klingenberg et al. [46]** | | | | |
| Inventory turnover with ROA | 11 | 170 | 0.09 | 0.0903 |
| Inventory turnover with ROE | 11 | 170 | −0.04 | −0.0401 |
| Inventory turnover with Asset Turnover | 11 | 170 | 0.66 | 0.6611 |
| Inventory turnover with Profit Margin | 11 | 170 | 0.13 | 0.1304 |

Source: Author's own.

Once we have the correlations and the sample sizes, we are prepared to calculate the individual effect size for each study and subsequently correct the measure for artifacts to ultimately obtain the global correlation coefficient. Before calculating the mean effect size, it is necessary to know the corrected correlation using the formula proposed by Zimmerman [48]. In addition to the formula, to obtain the corrected correlation, it will be necessary to use other correction factors, with complex mathematical formulation, which are included in the R-project package, allowing the calculation of the value $r_c$ (e.g., [49,50]).

The second step, after obtaining an unbiased and comparable correlation measure, is to ensure that each study contributes only one $r$ to the meta-analysis. Therefore, we will calculate the $r_{aggregated}$ using the method developed by Hunter and Schmidt [50], which considers the dependency between variables within the study, including the intercorrelation of effect sizes in the calculation. Since we do not have intercorrelations between effect sizes ($\bar{r}_{y_i y_j}$), we will use the value 0.5 in all cases, as recommended by Wampold et al. [51].

Table 5 shows the values after aggregating the effect measures according to Hunter and Schmidt's methodology [50,52]. In some cases, the $r_c$ is the same as in the previous table when a study provides only a single effect size.

**Table 5.** Corrected correlations and variance for each study.

| Publication | n | $r_c$ | Variance $r_c$ |
|---|---|---|---|
| Lawrence and Hottenstein [36] | 116 | 0.4215 | 0.005880342 |
| Jayaram and Vickery [37] | 49 | 0.1849 | 0.019433183 |
| Claycomb et al. [38] | 200 | 0.1604 | 0.004769878 |
| Fullerton and McWatters [40] | 91 | 0.3366 | 0.008735974 |
| Nahm et al. [41] | 224 | 0.4358 | 0.002942722 |
| Challis et al. [42] | 1024 | 0.2085 | 0.000894375 |
| Avittathur and Swamidass [43] | 26 | −0.2765 | 0.034117618 |
| Fullerton and Wempe [44] | 121 | 0.284 | 0.007043278 |
| Inman et al. [45] | 96 | 0.1973 | 0.009722745 |
| Klingenberg et al. [46] | 170 | 0.2662 | 0.005108264 |

Source: Author's own.

Now that we have obtained the effect sizes from each study, we are now ready to perform the actual meta-analytical calculation and integrate all the measures into one that serves to determine whether there is a relationship between financial performance and JIT.

The meta-analysis will be conducted following the methodology proposed by Cooper et al. [53], correcting only for sampling error. Therefore, our MA will be centered on sampling error and not comparative, since we do not have any control or placebo group. Given that the sample characteristics and methods used in the analyzed publications are not identical, we select a random-effects model. As outlined above, this model takes the form:

$$\theta = \mu + v_i^*$$

where $\theta$ represents the true effect size for each study $i$, $\mu$ is the true mean effect size, and $v_i^* = v_i + \tau^2$, which includes the variance of errors within and between studies. The model will be estimated using restricted maximum likelihood (REML), which is recommended according to the literature [54]. The weights used for calculation ($w_i$) are the inverse of the variance of sampling error for each study [55]. This method has the highest acceptance in the existing literature.

Regarding the calculation of the overall effect size, the open-source program R-Project v.4.2.3 has been used with various libraries available online:

- MAc library [56]: For aggregating measures at the study level, the agg function has been used, and the var_r function has been used to estimate its variance.
- Metafor library [57]: To estimate the different parameters of the random-effects model, the rma function has been used. Sensitivity analysis has been carried out using the leave1out function. The radial function has allowed generating the Galbraith plot (1994).
- Meta library [58]: The metabias function has been used to conduct the regression test to assess publication bias.
- Stats library [59]: The qqnorm function has allowed generating the QQ plot.

The result was obtained after entering the following variables and constraints into the software:

- Study-level effect sizes: data from corrected correlations (Table 5).
- Estimation model: random-effects model.
- Estimation method: "Restricted Maximum Likelihood".
- Weights $w_i$: inverse of the variance of sampling error for each study.

## 4. Results

The results of the meta-analysis of correlations to assess the association between the JIT construct and financial performance indicate that this association is significant and positive, with a value of 0.263 and a 95% confidence interval of (0.187–0.339).

The output from the statistical program is shown in Table 6, where *estimate* represents the overall correlation, *se* is the standard error of the correlation, *zval* is the z-statistic, *pval* is the *p*-value associated with the estimate, and *ci.lb* and *ci.ub* are the 95% confidence intervals.

**Table 6.** Output of estimation from R-Project.

| *estimate* | *se* | *zval* | *pval* | *ci.lb* | *ci.ub* |
|---|---|---|---|---|---|
| 0.2626 | 0.0386 | 6.8042 | <0.0001 | 0.1869 | 0.3382 |

Source: Author's own.

As observed, the confidence interval is very narrow and does not include zero as a value. According to Lipsey and Wilson [55], a correlation below 0.10 is low, around 0.25 is moderate, and above 0.31 is high. It can be inferred that we are facing a moderate relationship, since the value provided by the program is 0.2626. The confidence interval also confirms this, as the lower limit of the interval does not approach the values of 0.10, so it is not considered a low correlation, and the upper limit is above the high correlation value.

Figure 1 shows the estimated correlations ($r_c$) for each of the studies, along with their 95% confidence intervals (the thicker the point, the greater the precision of the correlation). The weights assigned to each of the studies are also displayed. At the bottom of the graph, the estimation of the overall correlation obtained through the random-effects model is shown. Now that the positive correlation is known, it is time to analyze the possible existence of heterogeneity among the variables. For this purpose, the R-Project provides a table with various indicators, which are expressed in Table 7.

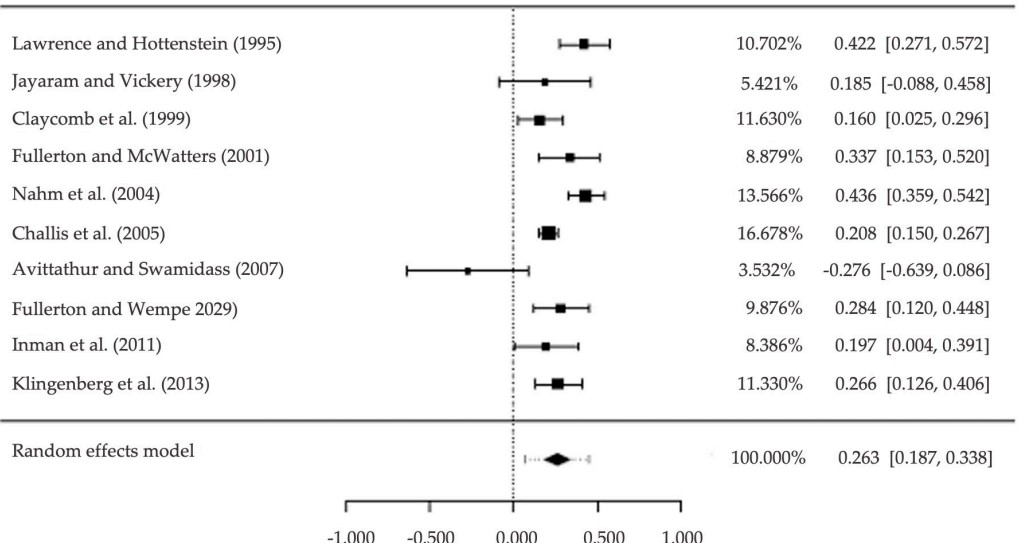

**Figure 1.** Plot of the individual effect measures, 95% confidence intervals, and weights assigned to each study, as well as the overall effect sizes. References in order of appearance in the figure: [36–38,40–46] Source: Author's own.

**Table 7.** Heterogeneity measures and 95% confidence intervals.

|  | **Estimation** | **Lower CI** | **Upper CI** |
|---|---|---|---|
| $\tau^2$ | 0.008 | 0.0029 | 0.1087 |
| $\tau$ | 0.0896 | 0.0542 | 0.3297 |
| $I^2(\%)$ | 62.3731 | 37.7836 | 95.7314 |

Source: Author's own.

The percentage of unexplained variance in the global effect size ($I^2$) is 62.37%, indicating moderate heterogeneity, although the upper limit of the confidence interval is very high, suggesting that much of the heterogeneity is due to real differences between studies rather than sampling error. $\tau$ represents variability not attributable to sampling error.

This source of heterogeneity needs to be sought among the characteristics and variables of the included studies, and a meta-regression with the candidate variables needs to be carried out. The last step before entering the interpretation phase of the results is to diagnose whether the results obtained in the meta-analysis are reliable or not. For this purpose, we will proceed to analyze publication bias, normality testing, and sensitivity analysis. Publication bias will be analyzed numerically using a regression test (Table 8) and graphically with the Galbraith plot [60] which can be observed in Figure 2.

**Table 8.** Publication bias.

| T-Statistic | Degrees of Freedom | *p*-Value |
|:---:|:---:|:---:|
| −0.1668 | 8 | 0.8717 |

Source: Author's own.

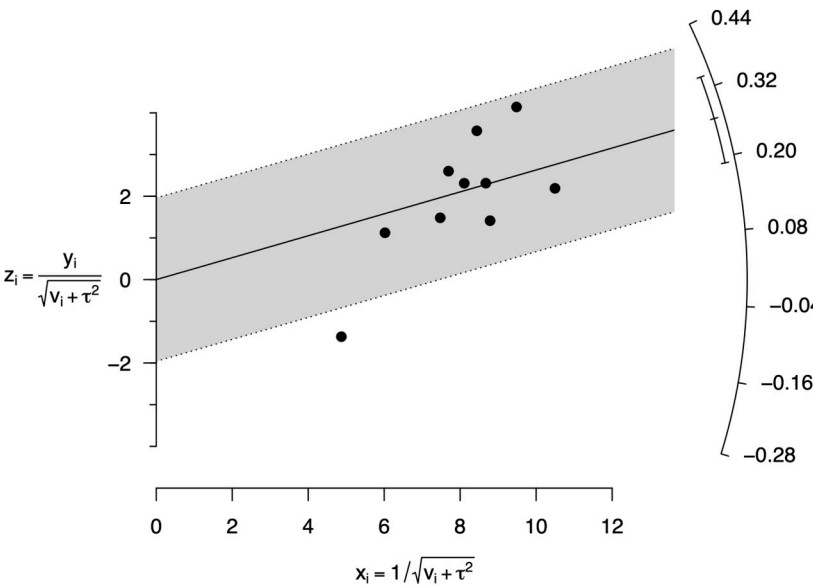

**Figure 2.** Galbraith plot. Source: Author's own.

The regression test to assess publication bias suggests that there is no publication bias. Based on the data, we can accept the null hypothesis in the hypothesis test, which tests whether there is publication bias or not, where the null hypothesis is that there is no publication bias.

On the other hand, in the Galbraith plot, it can be observed that all studies fall within the confidence interval and around the line, except one, Avittathur and Swamidass [43], which has a negative correlation and a higher standard deviation. Studying both the graphical and numerical evidence, therefore, it can be concluded that there is no publication bias.

To analyze the normality of the distribution, the graphical method of the Q-Q plot will be used. As can be observed in Figure 3, all correlations, except for the one from the study by Avittathur and Swamidass [43], are arranged around the line and within the confidence interval. We can assume normality in the distribution of the observed effect sizes. The confidence interval is indicated by the dashed line. Therefore, it appears that all studies, except for Avittathur and Swamidass [43], come from the same population.

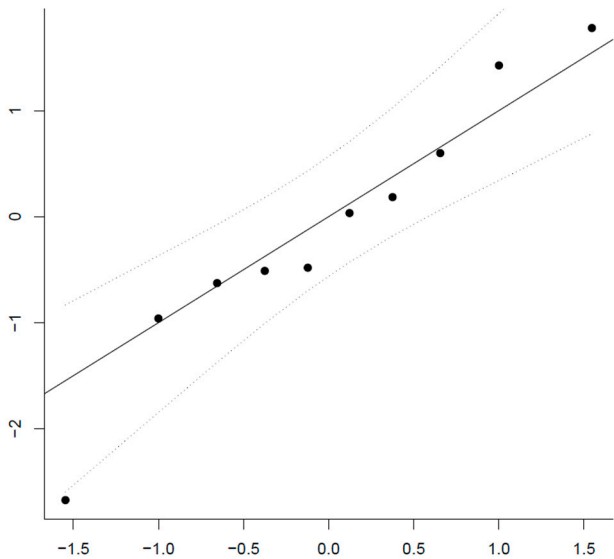

**Figure 3.** Q-Q plot (sample quantiles vs. theoretical quantiles). Source: Author's own.

Finally, the sensitivity analysis [28] will be conducted using the "leave-one-out" method, and the results are presented in Table 9, which shows that none of the publications have a significant influence on the overall result. The estimated overall correlations hardly change. However, it is observed that by removing Avittathur and Swamidass [43], the confidence interval of the effect size becomes narrower. On the other hand, it appears that Nahm et al. [41] is contributing to increasing heterogeneity, as when it is removed, $I^2$ decreases.

**Table 9.** Sensitivity analysis results.

| Publication Removed | $r_i$ | $se$ | $p$-Value | Lower CI ($r_i$) | Upper CI ($r_i$) | $\tau^2$ | $I^2$ |
|---|---|---|---|---|---|---|---|
| Lawrence and Hottenstein [36] | 0.2449 | 0.0391 | 0 | 0.1683 | 0.3216 | 0.007 | 58.9738 |
| Jayaram and Vickery [37] | 0.2666 | 0.0404 | 0 | 0.1874 | 0.3457 | 0.0085 | 65.585 |
| Claycomb et al. [38] | 0.2757 | 0.0414 | 0 | 0.1946 | 0.3568 | 0.0082 | 62.3992 |
| Fullerton and McWatters [40] | 0.2547 | 0.0412 | 0 | 0.1739 | 0.3354 | 0.0085 | 64.7077 |
| Nahm et al. [41] | 0.238 | 0.0344 | 0 | 0.1705 | 0.3055 | 0.004 | 42.2461 |
| Challis et al. [42] | 0.2679 | 0.0472 | 0 | 0.1754 | 0.3603 | 0.0116 | 62.6683 |
| Avittathur and Swamidass [43] | 0.2822 | 0.0344 | 0 | 0.2148 | 0.3495 | 0.0051 | 53.5539 |
| Fullerton and Wempe [44] | 0.2592 | 0.042 | 0 | 0.1769 | 0.3414 | 0.0089 | 65.2529 |
| Inman et al. [45] | 0.2679 | 0.0412 | 0 | 0.1871 | 0.3487 | 0.0086 | 65.1497 |
| Klingenberg et al. [46] | 0.2607 | 0.0427 | 0 | 0.1771 | 0.3444 | 0.0092 | 65.0392 |

Source: Author's own.

## 5. Discussion and Conclusions

The meta-analysis suggests the substantial impact of JIT practices on financial metrics, resolving ambiguities presented by individual studies with mixed outcomes. Thus, by analyzing correlation coefficients across multiple studies, we have established a robust positive relationship [42] between JIT practices and financial outcomes. This challenges traditional perceptions of JIT as solely an operational tool, positioning it as a strategic asset pivotal for achieving financial resilience and growth. The findings suggest that the strategic integration of JIT practices into broader organizational strategies can enhance financial performance, offering a competitive edge in today's volatile market environment. Our study therefore advances our understanding of the strategic implications of Just-In-Time (JIT) practices, presenting empirical evidence that JIT implementation is not only a driver of operational efficiency but also a critical contributor to financial health.

Building on the established operational efficiencies, this discussion delves deeper into how these benefits translate into tangible financial gains. JIT practices, characterized

by reducing inventory waste and streamlining production processes, directly decrease operational costs. These reductions can significantly enhance profitability and Return on Investment (ROI). For example, decreased cycle times and improved responsiveness to market demands not only increase customer satisfaction but can lead to higher sales volumes and better profit margins. The integration of JIT with strategic organizational practices such as quality management and supply chain optimization further amplifies these financial benefits, providing firms with a substantial competitive edge in today's dynamic market environment.

The observed variability across studies, especially those with wide confidence intervals, suggests a complex landscape where JIT's effects are not uniformly experienced across different organizational contexts. This complexity may lead to future lines of research that call for deeper exploratory efforts into the sources of heterogeneity through advanced analytical techniques such as meta-regression or subgroup analysis, aiming to uncover nuanced factors that influence the effectiveness of JIT practices.

In addition to internal operational improvements, the effectiveness of JIT practices is also shaped by external factors such as market volatility, economic cycles, and supply chain disruptions. During periods of economic growth, JIT enables organizations to efficiently scale operations to meet increased demand. Conversely, during downturns, the minimal buffers within JIT systems can expose firms to risks if not managed with strategic foresight. Supply chain disruptions, particularly in JIT systems that depend on the timely delivery of components, can significantly impact production continuity and, consequently, financial performance. We must therefore highlight the need for robust contingency strategies, such as diversified supplier bases or enhanced supplier collaborations, to mitigate such risks and ensure that JIT systems contribute positively to financial outcomes under varying economic conditions.

Furthermore, this meta-analysis also paves the way for future inquiries into lesser-studied areas, particularly the role of JIT in small enterprises. Despite their significant potential for benefiting from JIT methodologies, small enterprises have been relatively overlooked in the literature. Addressing this gap could reveal critical insights into how these organizations can navigate the unique challenges and opportunities presented by JIT practices, enriching both theoretical frameworks and practical applications in the field of operations management.

Additionally, our research can inspire the mechanisms through which JIT practices influence financial metrics, potentially bridging the gap between operations management and financial strategy, both in the medium and long term. By eliminating waste, optimizing processes, and improving supply chain efficiency, companies can achieve cost reductions, increases in product quality, and greater agility in responding to market demand. But there are also two key factors to consider: investment in technology and labor. On the one hand, investment in technology plays a crucial role. The acquisition of specialized systems and software for JIT implementation may require an initial investment, but in the long term, it can translate into significant savings and productivity improvements. Technology not only automates processes, but also improves production planning and optimizes inventory management, leading to greater operational efficiency and a better ability to adapt to changes in the environment. Thus, this study encourages a multidisciplinary approach to understanding JIT's role, urging scholars to consider its broader implications within strategic management frameworks.

In this sense, it is important to recognize the broader operational principles embedded within JIT that align with sustainable business practices. JIT's emphasis on minimizing waste, reducing inventory levels, and improving resource efficiency not only enhances operational efficiency but also supports environmental sustainability objectives indirectly. However, the studies included in our meta-analysis did not directly measure environmental outcomes such as carbon footprint reduction, energy savings, or other sustainability metrics. As such, while we propose that JIT practices may theoretically support sustainability goals, empirical studies focused on these specific metrics are required to draw definitive

conclusions. This gap presents a valuable opportunity for future research to integrate environmental performance measures into the assessment of JIT impacts, thereby providing a more comprehensive understanding of JIT's role in promoting sustainable manufacturing and operational practices.

Finally, from a practitioner's point of view, to effectively implement JIT, managers should ensure alignment with broader organizational strategies, tailoring JIT principles to specific company needs and industry conditions. Investment in relevant technologies is crucial for streamlining processes and improving real-time data flow across supply chains. Additionally, managers should focus on fostering strong supplier relationships and robust risk management strategies to mitigate potential supply chain disruptions inherent in JIT systems with minimal inventory buffers. Training programs are essential to cultivate a workforce proficient in JIT methodologies and committed to continuous improvement. Continuous monitoring and systematic analysis of JIT practices should be employed to refine processes and eliminate inefficiencies, thereby sustaining financial gains and competitive advantages. For sectors like automotive manufacturing, where precision and timing are critical, integrating JIT deeply into every stage of production can lead to significant cost savings and operational efficiencies. By adopting these focused strategies, managers can leverage JIT to not only improve operational efficiency but also achieve substantial financial and strategic benefits in dynamic market environments.

From a broader perspective, governments can enhance the adoption and effectiveness of Just-In-Time (JIT) practices through a series of focused public policies. Promoting JIT in small and medium enterprises via financial incentives such as tax breaks or technology upgrade subsidies can help these firms improve competitiveness and financial stability. Supporting innovation in JIT-related technologies through funding for research and development can spur advancements in automation and supply chain management systems. Furthermore, policies aimed at developing resilient supply chain infrastructures can mitigate risks associated with JIT's minimal inventory systems, particularly in volatile markets. Incorporating JIT principles into regulatory standards can also standardize practices across industries, ensuring that JIT adoption leads to substantial financial and operational improvements. Collectively, these policies not only encourage the widespread use of JIT but also align it with broader economic resilience and sustainability goals.

**Author Contributions:** Conceptualization, J.G.-C. and C.R.-G.; methodology, J.G.-C.; formal analysis, J.G.-C.; investigation, J.G.-C. and C.R.-G.; writing—review and editing, J.G.-C. and C.R.-G.; funding acquisition, J.G.-C. All authors have read and agreed to the published version of the manuscript.

**Funding:** Javier García-Cutrín acknowledges financial support from Xunta de Galicia (ED431B 2022/03).

**Institutional Review Board Statement:** Not applicable.

**Informed Consent Statement:** Not applicable.

**Data Availability Statement:** The data used to support the findings of this study can be made available by the corresponding author upon request.

**Conflicts of Interest:** The authors declare no conflict of interest.

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
