# Peer review of "Enhancing Corporate Sustainability through Just-In-Time (JIT) Practices: A Meta-Analytic Examination of Financial Performance Outcomes"

_sustainability, doi:10.3390/su16104025_

Round 1
Reviewer 1 Report
Comments and Suggestions for Authors
Review of “Enhancing Corporate Sustainability through Just-In-Time (JIT) Practices: A Meta-Analytic Examination of Financial Performance Outcomes”
This study consolidates correlation coefficients from 12 empirical studies using meta-analytic tests, demonstrating that the adoption of Just-In-Time (JIT) practices leads to enhanced financial performance. By revealing that JIT operational practices positively influence financial outcomes, the study makes a contribution to existing literature. However, there are some aspects that warrant consideration to strengthen the study's conclusions.
1. Number of papers analyzed
The study aims to draw comprehensive conclusions by conducting a meta-analysis of previous research on JIT and financial performance. While the search spanned 23 journals over nearly 30 years (1995 to 2022), only 12 studies were ultimately included in the analysis. The methodology for selecting these studies lacks explanation, and the number of papers analyzed is insufficient. This limitation must be addressed by broadening the scope of the literature reviewed and including more studies in the analysis.
2. Concreteness of analysis
A more detailed analysis is necessary to thoroughly investigate the findings of previous studies. Specifically, the study should clarify the perspectives from which financial performance was assessed, identify the financial performance proxies utilized, highlight cases where the relationship is more pronounced, and explore how external factors, such as environmental changes, may have influenced the results. While the paper establishes a link between JIT implementation and improved financial performance, a more granular analysis is essential for deeper insights.
3. Relationship with Sustainability
The abstract of the paper suggests a connection between JIT practices and sustainable business operations, including potential expansion into environmental sustainability initiatives. However, the study's findings may not directly support discussions on sustainability. Further exploration is required to examine how JIT adoption aligns with broader sustainability goals, necessitating additional investigation and discussion within the paper.
Reviewer 2 Report
Comments and Suggestions for Authors
The study "Enhancing Corporate Sustainability through Just-In-Time (JIT) Practices: A Meta-Analytic Examination of Financial Performance Outcomes" systematically examines the relationship between JIT practices and financial performance in manufacturing firms. The authors employ a comprehensive meta-analytical framework to synthesize findings from a broad spectrum of studies, aiming to resolve the variability in results and interpretations that have characterized previous research. I provide the following comments as constructive criticism for developing the manuscript:
The study draws from a relatively narrow range of empirical studies published between 1995 and 2022. This period covers significant technological and economic shifts which might affect the applicability of JIT practices. I suggest that the authors provide more justification for the sample period.
The literature review and theoretical perspective should be enhanced with more discussion and recent literature that has discussed the impact of sustainability on financial performance. I suggest that authors consider recent studies such as:
https://doi.org/10.1108/CG-03-2022-0105
The study reports a moderate positive correlation between JIT adoption and superior financial performance. However, the interpretation of this effect size could benefit from a deeper analysis of its practical significance in real-world settings. Moreover, the moderate heterogeneity observed suggests that the relationship may not be uniform across all contexts.
While the study bridges the gap between operational and financial performance, the discussion could be enriched by a more nuanced exploration of how operational benefits translate into financial gains.
The discussion could benefit from a more detailed consideration of external factors that influence the relationship between JIT practices and financial performance, such as market volatility, supply chain disruptions, and economic cycles.
The study makes significant contributions by consolidating the diverse body of research on JIT practices and financial performance. However, further elaboration on how this synthesis advances theoretical frameworks in operations management and finance would strengthen the paper. Specifically, integrating JIT within broader strategic management and sustainability frameworks could offer new theoretical insights.
The practical implications for managers are clearly highlighted, especially the strategic view of JIT as not just an operational tool but a financial performance enhancer. Expanding on this by offering more concrete guidelines or considerations for managers looking to implement JIT would make the study's contributions more actionable.
Comments on the Quality of English Language
Minor
Reviewer 3 Report
Comments and Suggestions for Authors
Dear authors, congratulations on the work presented.
This is an important area of study (financial impact), considering the contributions to companies' sustainability through JIT practices. In my opinion, the greatest merit of this work is that it highlights the need to study the sustainability of companies, through the prism of the effectiveness and efficiency of JIT practices. As mentioned in the article, this is a very complex reality to study, so the combination of different research methodologies, together with different organizational realities (multiple cases), could be the way so that in the future, the practical community can adopt, approximately, more financially sustainable management principles.
This is, therefore, a study that, in addition to the theoretical contribution, provides good reflection for company management.
I believe that it would be important to have space for a reflection on the way in which JIT practices impact the financial results of companies, also on an ongoing basis, in the medium and long term, relating this topic to investment in technology, with incentives on the part of public funds, with the motivational factors and productivity of workers, and with the confidence of businesspeople in the face of contexts of geopolitical and geostrategic uncertainty that occur in different global contexts.
In any case, we have a very good contribution. Congratulations and good luck for future work.
Round 2
Reviewer 1 Report
Comments and Suggestions for Authors
The manuscript has been adequately revised based on my previous comments.